# Corporate Governance and Stock Price Synchronicity: Empirical Evidence from Vietnam

**Anh Huu Nguyen** *[ID], **Thu Minh Thi Vu**[ID] **and Quynh Truc Thi Doan**

School of Accounting and Auditing, National Economics University, Hanoi 100000, Vietnam;
vuminhthu.neu@gmail.com (T.M.T.V.); dtquynh0711@gmail.com (Q.T.T.D.)
* Correspondence: nguyenhuuanh68@gmail.com or anhnh@neu.edu.vn; Tel.: +84-906163535

**Abstract:** This research is conducted to investigate the impact of corporate governance on stock price synchronicity in the context of the Vietnamese market. The paper tests four hypotheses proposing the effect of four crucial components of corporate governance including board size, board independence, managerial ownership, and foreign ownership on stock price synchronicity. The study sample includes 247 non-financial listed companies on the Ho Chi Minh Stock Exchange (HOSE) in Vietnam over a period of five years from 2014 to 2018. The fixed effects model is employed to address econometric issues and to improve the accuracy of the regression coefficients. The research results show the positive impact of board size and foreign ownership but the negative impact of managerial ownership on stock price synchronicity. This study confirms the viewpoint that stocks in the market move more together when the firms' corporate governance gets better. In other words, the research findings suggest that low synchronicity signifies the corporate intransparency and weak information environment and vice versa. From this, the paper provides a new insight to managers on how to improve stock price synchronicity with corporate governance.

**Keywords:** corporate governance; stock price synchronicity; firm-specific information; fixed effects model; Ho Chi Minh stock exchange (HOSE); information environment

**JEL Classification:** G30; G32; M40

## 1. Introduction

The relationship between stock price synchronicity and price informativeness has been pointed out to be closely related to the quality of the information environment and the efficiency of the capital market. Stock price synchronicity measures the degree to which individual stocks co-move with the market, reflecting the amount of systematic volatility relative to idiosyncratic volatility (or total volatility). After Roll (1988) suggested the market model that formulated relations between stock price synchronicity and firm-specific information, the conventional wisdom found in the following studies is that low level of synchronicity is the signal of more firm-specific information incorporated in stock price thanks to the transparent information environment both inside and outside firms.

However, recently, there is an increasing number of researches that document the opposite result. Especially, most of these researches were carried out in emerging markets, and they reached the same conclusion that more efficient markets and effective corporate mechanisms make stock prices move simultaneously, resulting in a high price synchronicity.

In an efficient market, firm's stock prices move to reflect available information (both firm-specific and market-wide information), and they respond only to announcements that are not already predicted. When the information environment surrounding a firm improves and more firm fundamentals are available, participants in the market are able to make better forecasts about the occurrence of the

firm's future events. Accordingly, Morck et al. (2000) proposed that strong investor protection in the developed markets encourages investors to make informed trading, leading to a high level of private information in stock price. However, in the less efficient markets, such as emerging markets, the typical firm characteristic of high concentrated ownership by family members or government enables them to withhold firm private information to outside investors, exacerbating the information asymmetries and discouraging investors to make informed trading. Therefore, it is more interesting for researchers to investigate the relationship between information environment and stock price synchronicity in the emerging markets because stock prices track their fundamentals more loosely in a less efficient and speculative market.

The first stock exchange in Vietnam was introduced in 2000 with four listed firms. After 18 years, we have made rapid growth with two stock exchanges operating and more than 700 firms listed (Nguyen et al. 2019). This achievement follows the economic growth as the whole and the improvement of the market efficiency through the period (Nguyen and Hanh 2012). Research of Truong et al. (2010), employing data from 2000 to 2004, failed to find support for weak-form efficiency in the case of Ho Chi Minh Stock Exchange. Duc et al. (2011) also suspected the efficiency of the Vietnamese market under the view of investors' behavioral finance. Yet, three years later, Gupta et al. (2014) tested daily, weekly, monthly, and quarterly data, and came to the conclusion that the Vietnamese stock market is weak-form efficient, at least in the later period. Especially, recent research of Dzung and Quang (2019) applying the new concept "Adaptive market hypothesis", which is the harmonization of efficient market hypothesis and behavioral finance, concluded that Vietnamese stock market's behaviors have gradually improved and are in line with the adaptive market hypothesis. Besides these developments, we cannot deny that the Vietnamese stock market is still a small frontier or emerging market with low liquidity compared to other stock markets. Weak corporate governance with high ownership concentration by family or government and low foreign ownership are well-characterized by the market. For many years, the Vietnamese government imposed restrictions on foreign ownership in domestically listed firms: Up to 49% of the equity for the listed companies and up to 30% for the listed banks. In addition, with regard to the macro-economic context, the government still adopts policies that result in less transparency and an inefficient environment, providing poor protection for public investors from corporate insiders.

This research is motivated by the characteristics of the emerging market in Vietnam and the recent debate on stock price synchronicity. Our research focuses more on corporate governance as the crucial mechanism that affects the transparency of firms' inside information environment in the context of Vietnamese emerging market. The result of research reveals that low stock price synchronicity in Vietnamese stock market is associated with ineffective corporate governance system, which limits transparency to the market forces. This finding contributes to the growing literature by adding more support to the view that a weak information environment leads to low synchronicity and vice versa. It also explained why many prior studies employed stock price synchronicity as the direct measure rather than the inverse measure for stock price informativeness. Furthermore, from our perspective, this result might be inferred to other emerging markets, but not developed markets because we posit that market participants behave and interact very differently in developed markets.

This paper consists of five sections. Following the Introduction section is the Literature Review and Hypotheses Development. Section 3 describes data collection, variables, and regression model. The next part discusses empirical results, and the last one concludes main ideas, referring to implications for managers of Vietnamese companies and government authority.

## 2. Literature Review and Hypotheses Development

Stock price synchronicity was first proposed by Roll (1988) in his studies about firm-specific information or investors' private information impounded in stock price. His study relied on the traditional capital asset pricing model (CAPM) and suggested that stock return' variation can be divided into two different variations: The first one is the systematic variation including market and industry variation, and the second one is variation related to firm-specific information. The first variation called

stock price synchronicity is captured by the value of R-square (coefficient of determination) in the model. Roll (1988) also stated that stock price's co-movements depend on the amount of firm-level and market-level information incorporated into stock prices.

From this, R-square has been widely used on a large body of literature, both empirical and theoretical, as the direct measure for stock price synchronicity or indirect measure for firm-specific information in the stock price. However, unidentified implications of R-square has caused two dramatically different views of what is actually reflected by the stock price synchronicity and its relation to the information environment which comprises of two levels: Market-level (e.g., property rights protections and government index) and firm-level (e.g., corporate governance mechanism, reporting, and disclosure). It is well-established that both general information environment and firm's corporate governance in developed markets are more efficient and transparent than those in emerging markets (Chen and Yuan 2004; Jin and Myers 2006; Hu and Liu 2013). Accordingly, a school of thought proposed that strong investor protection in developed markets encourages investors to make informed trading, leading to a high level of private information in stock price and, therefore, lower synchronicity (Morck et al. 2000; Bertrand et al. 2002; Fernandes and Ferreira 2009).

However, another strand of literature questions such a negative relationship and argues that lower synchronicity indicates poor information environment as well as inefficient corporate governance and vice versa. There is not a single study affirming the positive relationship between efficient information environment and stock price synchronicity. Dasgupta et al. (2010) documented this argument by performing both theory and empirical research. He examined the impact of both time-variant and time-invariant information disclosure and came to the consistent conclusion that the improvements of transparency in the investment environment can increase R-square. Especially, this standpoint seems to be more dominant in the studies over the emerging markets. Piotroski and Roulstone (2004) and Chan and Hameed (2006) proved that in emerging markets, more analyst coverage means more information interpreted for investors and higher stock price synchronicity. Our research supports this point of view regarding the emerging market environment in Vietnam. We focus more on the impact of corporate governance on the co-movement of the firm's stock return with the market's rather than the overall market environment.

Corporate governance is the way of directing and controlling a company. Among related theories, agency theory is the root cause of the appearance and development of corporate governance mechanism. Though corporate governance can be approached through different perspectives, from operational (narrow) to stakeholder (wide) view, central among the concept is the shareholders, directors, and management's thinking and practicing. The major discussion over corporate governance in literature covers topics of board structure (the size, components, and duties of the board). Therefore, a good corporate governance is a good combination and arrangement of the board structure's elements in order to effectively resolve agency conflict as well as reduce information asymmetry.

We totally agree with the firms' common corporate governance features in emerging markets discussed above (e.g., higher ownership concentration by family or state and less transparent disclosure compared to firms in developed markets), which are also characterized by firms in Vietnam. However, we argue that a low level of stock return's co-movement with the market's return associates with weak corporate governance mechanism of firms in emerging market and vice versa for the following reasons:

Firstly, good corporate governance means more transparent and reliable information disclosed by firm that will help to increase the accuracy of investor forecasts. Herawaty and Solihah (2019) proposed that the better of corporate governance's implementation enables companies to reduce earnings manipulation that management is often tempted to do. According to Jin and Myers (2006), information environments that are more transparent reveal more private information about firms to outside investors, leading to the improvement in the investor's anticipations. Investors then, therefore, trade stocks based on these predictions and incorporate the likelihood of predicted events' occurrence into stock price. As a result, if these events occur in the future, they will not create much surprise and shock, leading to little reaction from the market. This mechanism was described by Dasgupta

et al. (2010) as "the intuitive implication of market efficiency" because stock price responds only to announcements that are unknown or not already predicted by the market's participants. A more informative stock price today should be associated with less firm fundamentals incorporated in this stock price in the future and higher synchronicity, providing that the synchronicity could consistently measure the amount of private information in stock return.

Secondly, our literature review shows foreign and institutional ownership can effectively monitor and improve firm's corporate governance (Chung and Zhang 2011; He et al. 2013). Especially, many studies documented that these two kinds of outside investors are interested more in stocks of companies with better managerial performance and better disclosure (Giannetti and Simonov 2006; Parrino et al. 2003). Farooq and Ahmed (2014) explained that as the most diversified investors in the market, institutional investors, minimize risk of firm-specific information and expect to experience only wide-market risk. Hence, the variation of stock return could be mainly explained by the variation of market return, signifying high stock return co-movement with the market or high stock price synchronicity as Hu and Liu (2013) confirmed "we do find stocks with higher R-square are more likely to be widely held by institutional investors than those with lower R-square."

Last but not least, according to the findings of researchers who concluded on the negative relationship between stock return synchronicity and corporate mechanism, the values of R-square in developing or emerging markets are often higher than in developed markets because of the lesser firm-specific information in stock price in developing and emerging markets. However, when calculating R-square in the Vietnamese market, we received a quite low value of average R-square for firms. This result is consistent with the findings in some of the recent researches that low synchronicity is one of the firm characteristics in emerging markets, showing a poor information environment (Li et al. 2014; Kelly 2014).

*2.1. Board Size and Stock Price Synchronicity*

The literature provides mixed results on the relationship between board size and corporate mechanism. While Jensen (1993) noted that smaller size of the board is more effective, Yermack (1996) and Xie et al. (2003) argued that larger size of the board could improve decision-making and monitor overall activities in a better way. We support the latter point of view as a large board associates with better corporate governance because it is less likely to be dominated by management and, thus, decreases asymmetric information, protecting shareholders' interests (Zahra and Pearce 1989). Our hypotheses assume the positive relationship between board size and stock price synchronicity.

**Hypothesis 1 (H1).** *Board size is positively related to stock price synchronicity.*

*2.2. Board Independence and Stock Price Synchronicity*

Banerji (2017) stated that in emerging economies, the firm's independent directors whose roles are mandated and protected by laws and regulations have played a significant role in the corporate mechanism. Findings of Dechow et al. (1996) suggested the negative relationship between board independence and the occurrence of financial statement fraud. Independence directors play a constraining role in real earnings management and encourage voluntary disclosure (Chouaibi et al. 2018). Furthermore, previous studies also proved that firms, which had a certain minimum number of independent directors, had relatively higher market valuation (Ammann et al. 2011). In brief, it can be proposed that independent directors are more likely to monitor effectively and help to improve the quality and reliability of firm disclosure, resulting in better information environment in companies.

**Hypothesis 2 (H2).** *Board independence is positively associated with stock price synchronicity.*

*2.3. Managerial Ownership and Stock Price Synchronicity*

Managerial ownership represents the portion of firm's shares owned by directors, their spouses, and children. The common thought is if stocks are held by directors in their firm, the less likely that they will invest in ineffective projects or consume additional resources. Therefore, managerial ownership is supposed to decrease agency conflict and is one of the most crucial corporate mechanism's components. However, as one corporate governance feature of emerging markets, high ownership concentration allows managers and controlling shareholders to evade effective information disclosure (Leuz et al. 2003). When managers hold a large share proportion of the companies, they are more influential and can be more self-interested (Fama and Jensen 1983). As a result, information asymmetries between insiders and outsiders are exacerbated by poor disclosure and agency problems become more serious. Therefore, we argue that firms with a large proportion of shares owned by managers have low stock price synchronicity.

**Hypothesis 3 (H3).** *Managerial ownership is negatively associated with stock price synchronicity.*

*2.4. Foreign Ownership and Stock Price Synchronicity*

The study of Vo (2017) employing synchronicity as the measure for stock price informativeness found the positive and significant relationship between foreign ownership and stock price informativeness measured by stock price synchronicity. He concluded that in the Vietnamese stock market, foreign ownership promotes more public information and reduces firm-specific information in stock price. Although there exist ongoing arguments on what is the stock price informativeness's effective measure, most of the authors agreed on the positive role of foreign ownership on corporate mechanism and disclosure quality improvement. Kho et al. (2009) explained that foreign shareholders who come from "good governance" countries to invest in "poor governance" countries could be more effective in monitoring managers and limiting agency problems.

**Hypothesis 4 (H4).** *Foreign ownership is positively associated with stock price synchronicity.*

### 3. Research Design and Data Collection

To investigate and explain the relationship between corporate governance and stock price synchronicity, we employ quantitative research and a deductive approach. There are four crucial elements of corporate governance tested, including board size, board independence, managerial ownership, and foreign ownership. The following model is proposed to test the hypotheses:

$$\text{SYNCH}_{it} = \beta 0 + \beta 1\,\text{BOARD}_{it} + \beta 2\,\text{INDEP}_{it} + \beta 3\,\text{MOWN}_{it} + \beta 4\,\text{FOREIG}_{it} + \beta 5\,\text{SOWN}_{it} + \\ \beta 6\,\text{SIZE}_{it} + \beta 7\,\text{AGE}_{it} + \beta 8\,\text{RISK}_{it} + \beta 9\,\text{ROE}_{it} + \beta 10\,\text{LEV}_{it} + \beta 11\,\text{MTB}_{it} + \varepsilon_{it} \tag{1}$$

where i indexes firms and t indexes years. Illustrations of variables and their codes are shown in the Table 1.

At first, we are based on original Roll's model to measure for stock price non-synchronicity. It is the model used on a large body of literature, both empirical and theoretical. Roll (1988) suggested that variation of a stock return can be decomposed into three different components: Variation related to market, variation related to the industry, and variation related to firm specification. Synchronicity is captured by the first two components that measure systematic variations. It can be estimated by R-square, where R-square is the coefficient of determination from the following regression:

$$R_{i,j,t} = \beta_{i,0} + \beta_{i,m}\,r_{m,t} + \beta_{i,j}\,r_{j,t} + \varepsilon_{i,t} \tag{2}$$

where:

$R_{i,j,t}$ is the return of firm i in industry j at time t

$r_{m,t}$ is the market return at time t

$r_{j,t}$ is the return of industry j at time t

**Table 1.** Illustration of variables.

| No. | Coding | Variables | Definition |
|---|---|---|---|
| 1 | SYNCH | Synchronicity | Estimated by Equations (3) and (4) |
| 2 | BOARD | Board size | Number of directors in the board |
| 3 | INDEP | Board independence | Number of independent directors in the board |
| 4 | MOWN | Managerial ownership | Percentage of share owned by directors, their spouses and children |
| 5 | FOREIG | Foreign ownership | Percentage of share owned by foreign investors |
| 6 | SOWN | State ownership | Percentage of share owned by the State |
| 7 | SIZE | Firm size | Total assets |
| 8 | AGE | Firm age | The firm age since the initial creation |
| 9 | RISK | Risk | The standard deviation of the firm's stock price |
| 10 | ROE | Profitability | Profit before tax divided by total equity |
| 11 | LEV | Leverage | Long-term debt to equity ratio |
| 12 | MTB | Market-to-book ratio | Book value of debts plus market value of equity divided by total value of assets |

However, there are two issues needed to be considered with the traditional capital asset pricing model suggested by Roll (1988) to compute synchronicity:

Firstly, in an emerging market, few industries can be more dominated in the economy than others, resulting in the difficulty to separate these industries' effect from the effect of the market. In addition, industry returns calculated from the few companies may reflect the company's specific news rather than industry news. Therefore, adding industry return in Equation (2) to estimate synchronicity can cause spurious results.

Secondly, because the value of $R^2$ is naturally bounded within the unit interval [0, 1], it may not serve as an appropriate dependent variable. Morck et al. (2000) suggested using unbounded logarithmic transformation of $R^2/(1 - R^2)$ in order to yield a dependent variable with a more normal distribution.

Thus, we finally follow the research of Morck et al. (2000) to estimate synchronicity:

+ Computing the $R^2$ from the following regression:

$$R_{i,t} = \beta_{i,0} + \beta_{i,m}\, r_{m,t} + \varepsilon_{i,t} \tag{3}$$

+ Then, synchronicity can be defined as:

$$\text{SYNCH}_{i,t} = \text{Log}\left[\frac{R^2}{1 - R^2}\right] \tag{4}$$

Our data set were collected for public firms listed on the Ho Chi Minh City Stock Exchange (HOSE) in the period of five years from 2014 to 2018. Firms included in the data sample must be listed and remain being listed during this period. We excluded data of banks, insurance, and other financial companies from the data set due to their special business nature and financial behaviors. In addition, some invalid or data-missing observations were also eliminated. Eventually, the last sample contains 243 companies coming from 10 different industries with 1121 observations.

The data set includes both listed firms' accounting and market information on HOSE. Data were collected from Viet Stock Database, which belongs to Tai Viet Corporation, a leading financial information and data supplier in Vietnam.

For synchronicity, at first, we estimate the synchronicity regression (Equation (3)) for every firm during each year. Then, Synch for each firm is computed for each year in the sample (Equation (4)).

## 4. Empirical Results and Discussion

Table 2 summarizes the data's statistics for the study's sample over the period observed from 2014 to 2018, including maximum, minimum figures, mean, and standard deviation of both dependent, independent, and control variables.

In the period from 2014 to 2018, the lowest R-square value of listed firms in HOSE is 0.001, the highest is 0.93, and the average level is 0.084. The figures for synchronicity (SYNCH) are −3, 1.123, and −1.426, respectively. The values for these variables in the Vietnamese market are relatively low compared to those of China's market, with the average figure of R-square and SYNCH being 0.434 and −0.232, respectively (Gul et al. 2010). According to Kelly (2014), low synchronicity is one of the firm characteristics in an emerging market, showing a poor information environment. Vo (2017) also estimated R-square value for HOSE in the period from 2007 to 2015 and presented the result for the highest, lowest, and average value of SYNCH as −17.73, 0.94, and −2.27, respectively. The reason for this lower synchronicity in the period from 2007 to 2015 could be because of the effect of the crisis suffered by the Vietnam stock market in 2008. From a peak in 2007, Vietnam's stock market bubble burst and bottomed up in the period from 2008 to 2009. In 2008, the market fell 66%, from 921 to 316. From the world's third best performer in 2006, Vietnam became the third-worst-performing market in the world in 2011.

**Table 2.** Descriptive statistics.

| Variable | N | Minimum | Maximum | Mean | Std. Deviation |
|---|---|---|---|---|---|
| BOARD | 1121 | 3 | 11 | 5.65 | 1.526 |
| INDEP | 1121 | 0 | 10 | 3.83 | 1.493 |
| FOREIG | 1121 | 0.00 | 77.60 | 15.41 | 16.142 |
| MOWN | 1121 | 0.00 | 73.12 | 15.69 | 11.460 |
| SOWN | 1121 | 0 | 97 | 15.25 | 23.743 |
| SIZE * | 1121 | 128,013 | 287,974,176 | 4,430,180 | 14,562,579 |
| AGE ** | 1121 | 6 | 91 | 25.38 | 13.453 |
| RISK | 1121 | 103.153 | 52,852 | 3530 | 4746 |
| ROE | 1121 | −1.899 | 1.607 | 0.124 | 0.178 |
| LEV | 1121 | 0.0058 | 140.258 | 1.680 | 4.716 |
| MTB | 1121 | 0.094 | 9.043 | 1.131 | 0.607 |
| R-SQUARE | 1121 | 0.001 | 0.930 | 0.084 | 0.109 |
| SYNCH | 1121 | −3.000 | 1.123 | −1.426 | 0.777 |
| Valid N (listwise) | 1121 | | | | |

Source: Analyzed by authors. * SIZE unit: Million VND. ** AGE unit: Year.

BOARD represents the executive and non-executive directors' number in the board. The board size of non-financial firms listed on HOSE ranges from 3 to 11, and on average, there are nearly 6 board members. According to Vietnamese Corporate Law 2014, board size must be at least 3 and no higher than 11 members. The Vietnamese firms' average board size is lower compared to the average of 10 board members of firms in Spain, and 9 members in the U.S (Granado-Peiró and López-Gracia 2017; Kieschnick and Moussawi 2018).

Board independence (INDEP) indicates the number of independent or non–executive members in the board. HOSE listed firms have a maximum of 10 independent board members with an average of 4 members. Specially, there are firms that do not have any independent members in their board, which does not meet Vietnamese law's requirement for public companies stipulating that at least one-third of the total members in the Board of Directors are independent. It is a sign of the under the enforcement of Vietnamese law for stock market in practice.

For foreign ownership (FOREIG), the proportion is from 0% to 77.6%. The average percentage is 15.41%, with the high standard deviation of 16.1, which represents the big gap and difference between each firm's foreign ownership rate and the overall-market average figure. Foreign ownership was

lower for the previous period from 2007 to 2015, with the maximum and average ratio being 49% and 8.78%, respectively, Vo (2017). The reason for this difference is the issuance of Government Decree 60/2015 that removed the limit on listed companies' foreign ownership. The new decree lifted foreign ownership from 49% to 100 % for most sectors. As the Vietnamese stock market is becoming more and more attractive to foreign investors, the expectation is that the more considerable amount of foreign capital will be flow in the stock market of Vietnam in the following years.

The managerial ownership (MOWN) variable received value from 0% to 73.12% with an average percentage of 15.72% for Vietnamese listed firms. Accordingly, while some of the directors do not even own any shares of their companies, some in other firms hold a relatively high share proportion in their firm. This average level of managerial ownership is higher than that of Spanish firms, which is in the same level with firms in other countries (Granado-Peiró and López-Gracia 2017). The high value of MOWN standard deviation of 11.46 shows the big gaps and differences between the managerial ownership rates of companies and their average figure.

Table 3 represents the result of the correlation's matrix among variables for the sample of 1121 observations. It can be seen that the correlation coefficients are generally below 0.50, the only correlation that might be suspected is between profitability and market-to-book variable (correlation coefficient = 0.537). However, according to the variance inflation factors (VIF) presented with the regression result, we can conclude that there does not exist an issue of serious multi-co linearity among variables. Noticeably, board size (BOARD) and FOREIG are recognized to have the strongest positive correlation with stock price SYNCH, with the Pearson correlation of 0.449 and 0.452, respectively.

**Table 3.** Correlations between variables.

|  | SYNCH | BOARD | INDEP | FOREIG | MOWN | SOWN | SIZE | AGE | RISK | ROE | LEV | MTB |
|---|---|---|---|---|---|---|---|---|---|---|---|---|
| SYNCH | 1 | | | | | | | | | | | |
| BOARD | 0.449 ** | 1 | | | | | | | | | | |
| INDEP | 0.172 ** | 0.387 ** | 1 | | | | | | | | | |
| FOREIG | 0.452 ** | 0.440 ** | 0.207 ** | 1 | | | | | | | | |
| MOWN | −0.126 ** | −0.084 ** | −0.216 ** | −0.093 ** | 1 | | | | | | | |
| SOWN | −0.007 | −0.086 ** | −0.081 ** | −0.034 | −0.170 ** | 1 | | | | | | |
| SIZE | 0.197 ** | 0.232 ** | 0.252 ** | 0.184 ** | −0.078 ** | −0.008 | 1 | | | | | |
| AGE | −0.062 * | −0.016 | −0.043 | −0.103 ** | −0.04 | 0.138 ** | −0.084 ** | 1 | | | | |
| RISK | 0.038 | 0.038 | 0.123 ** | 0.204 ** | −0.006 | −0.018 | 0.152 ** | 0.104 ** | 1 | | | |
| ROE | 0.012 | 0.049 | 0.079 ** | 0.086 ** | 0.018 | 0.015 | 0.03 | 0.04 | 0.329 ** | 1 | | |
| LEV | 0.004 | −0.023 | −0.058 * | −0.083 ** | 0.011 | −0.042 | 0.017 | −0.003 | −0.038 | −0.403 ** | 1 | |
| MTB | 0.007 | 0.042 | 0.165 ** | 0.182 ** | −0.017 | 0.101 ** | 0.148 ** | 0.104 ** | 0.559 ** | 0.360 ** | −0.039 | 1 |

Source: Analyzed by authors. * Correlation is significant at the 0.05 level (2-tailed). ** Correlation is significant at the 0.01 level (2-tailed).

Also, based on each of the independent variables, we divided the firms into two subgroups according to the variable average value to observe the differences in R-square (stock price synchronicity) between the firms' subsample. The results are presented in Table 4. For BOARD's subgroups, the Min, Max, and Mean values of R-square are all higher in the group that has from seven members in the board than in the other group that has less than seven members. FOREIGN's subsamples show the similar trend while the opposite can be seen in the MOWN's subgroups.

We carry out F-test to make the decision between the pooled regression and the fixed effects model. The F-test = 3.37 with significant $p < 0.01$, signaling the better fit of the fixed model compared to the pooled regression model. Following, Hausman test was performed to decide between using the fixed effects model and the random effects model. The probability of Wald yielded in the test is less than 0.01, which indicates that the fixed effect model is better off for analysis purpose. Then, the diagnostic test is run and detects the existence of heteroskedasticity (conditional unequal variance) with $p$-value $< 0.01$. Therefore, we cluster (robust) the standard errors by the firm to deal with this problem.

Table 5 shows the outcomes for the fixed effects model (FEM), including firm fixed effect and robust FEM model. In these models, the coefficient of determination is 0.2968, meaning that 29.68% of the variation in stock price synchronicity can be explained by the independent variables.

**Table 4.** Descriptive statistics of R-square according to different subgroups.

| | | BOARD | | INDEP | | FOREIGN | | MOWN | |
| --- | --- | --- | --- | --- | --- | --- | --- | --- | --- |
| | | <7 Members | ≥7 Members | < 4 Members | ≥4 Members | <15% | ≥15% | <16% | ≥16% |
| R-Square | *Min* | 0.001 | 0.073 | 0.001 | 0.001 | 0.001 | 0.053 | 0.001 | 0.001 |
| | *Max* | 0.675 | 0.930 | 0.705 | 0.930 | 0.413 | 0.930 | 0.930 | 0.671 |
| | *Mean* | 0.040 | 0.177 | 0.087 | 0.073 | 0.046 | 0.161 | 0.119 | 0.029 |
| | *Median* | 0.029 | 0.113 | 0.031 | 0.064 | 0.015 | 0.104 | 0.141 | 0.018 |
| *Number of Observations.* | | 760 | 361 | 854 | 267 | 752 | 369 | 663 | 548 |

Source: Analyzed by authors.

**Table 5.** Regression results.

| | Fixed Effect | | Robust (Fixed Effect) | | VIF |
| --- | --- | --- | --- | --- | --- |
| | Coef. | Std. Err. | Coef. | Std. Err. | |
| BOARD | 0.1262 *** | 0.0264 | 0.1262 *** | 0.0276 | 1.49 |
| INDEP | −0.0220 | 0.0255 | −0.0220 | 0.0263 | 1.37 |
| FOREIG | 0.0183 *** | 0.0026 | 0.0183 *** | 0.0027 | 1.38 |
| MOWN | −0.0094 *** | 0.0028 | −0.0094 *** | 0.0032 | 1.12 |
| SOWN | −0.0012 | 0.0016 | −0.0012 ** | 0.0017 | 1.11 |
| SIZE | 0.1018 | 0.0000 | 0.1018 | 0.0000 | 1.14 |
| AGE | 0.0997 *** | 0.0127 | 0.0997 *** | 0.0127 | 1.07 |
| RISK | 0.0131 | 0.0188 | 0.0131 | 0.0175 | 1.62 |
| ROE | 0.0898 | 0.1331 | 0.0898 | 0.1173 | 1.47 |
| LEV | 0.0142 | 0.0152 | 0.0142 | 0.0145 | 3.82 |
| MTB | −0.1620 *** | 0.1145 | −0.1620 *** | 0.1034 | 3.95 |
| R-Square | 0.2968 | | 0.2968 | | |
| F-test | | | 3.37 *** | | |
| Hausman Test | | | 79.12 *** | | |
| Modified Wald (χ2) (Heterokedasticity) | | | $7.1 \times 10^{31}$ *** | | |
| Observations | 1121 | | 1121 | | |

Source: Analyzed by authors. ** Significance is at the 0.05 level (2-tailed). *** Significance is at the 0.01 level (3-tailed).

Generally, according the result revealed by robust FEM model, BOARD, FOREIG, firm size (SIZE), firm age (AGE), risk (RISK), profitability (ROE), and leverage (LEV) have positive effect on stock price SYNCH. In contrast, board independence (INDEP), MOWN, state ownership (SOWN) and market-to-book (MTB) show negative influence on stock price SYNCH. However, only six among independent variables show significant results.

Board size shows the positive and significant impact on stock price synchronicity under the FEM model. The higher the number of members in the board is, the higher chance that stock return will co-move with the wide market's return. Hypothesis H1 is accepted. A number of previous studies emphasized the importance and positive role of board size to the corporate mechanism. According to Anderson et al. (2004), a larger board can more effectively monitor financial reporting because it has an inverse relationship with the cost of debt. Research of Ntow-Gyamfi et al. (2015) explained that a firm with larger board size can increase its transparency, leading to the higher levels of stock co-movement with the market or synchronicity.

Similarly, foreign ownership is found to be positively associated with stock price synchronicity with the coefficient of +0.018 and *p*-value < 0.01. We accept the second hypothesis H2, that larger holding shares of firms' foreign investors will make their stock price move together. Among corporate governance's components that can have an impact on stock price synchronicity, foreigner's holding received a lot of attention from researchers. Although several studies interpreted the significant and negative relation between foreign ownership and synchronous characteristics of stock price, not only one research insisted on the opposite result. The finding of this study verifies the latter. In the

market with less transparent information environment, foreign investors feel more confident with their portfolios consisting of stocks from firms with good corporate governance practices. They, in turn, could exert pressure on both the external and internal information environment by installing high standards of reporting and disclosure. Research of Vo (2017) also postulated the similar outcome between foreign ownership and stock price synchronicity when he studied Vietnamese market in the past.

On the contrary, board independent is negatively but insignificantly correlated with the synchronous level of firm's stock return to the market's under both fixed effects and robust model. The Hypothesis H3 is rejected. As mentioned above, the number of independent board members in many firms listed on HOSE is not satisfied Vietnamese law for public companies and has not received enough attention from authorities. The cause of this incompliance might be because of the lack of both firms and regulators' understanding about the essential role of independent members to the company's performance as well as right protection for investors in the market, leading to the ineffective performance of independent board members in corporate governance.

As expected, managerial ownership is inversely related to synchronicity with the coefficient of −0.009 and *p*-value < 0.01 as shown in the fixed effects model's result. This significant correlation confirms the hypothesis H4. The larger amount of shares the managers own, the lower chance that stock return is synchronous with the wide-market return. The high level of managerial ownership is one of the corporate governance's typical characteristics of the emerging market in Vietnam. Research of Farooq and Ahmed (2014) asserted that ownership concentration is associated with the higher information asymmetries and the lower institutional holding. Owning a large amount of shares creates opportunities for managers to extract private control benefits at the expenses of outsiders (Bertrand et al. 2002). Consequently, information asymmetry increases and discourages outside investors including institutional investor to make informed trading and invest into the companies.

## 5. Conclusions

This paper examines the link between stock return synchronicity and corporate governance in the context of Vietnamese emerging market. The research found the positive impact of board size and foreign ownership but the negative impact of managerial ownership on stock price synchronicity in the context of an emerging market in Vietnam. Our study confirms the view that stocks in the market move more together when their corporate governance improves. Overall, this contributes to the growing body of literature by adding more support to the belief that higher synchronicity signifies the better quality of information environment. Accordingly, to improve stock price synchronicity, government authority and company managers should make an effort to establish a more transparent and information environment surrounding firms.

Our finding does not negate the opposite view proposed in previous papers on the negative relationship between synchronicity and corporate governance. Instead, it might suggest the different implications of R-square in different market conditions as the dynamic response of stock return synchronicity to changes in the information environment. Therefore, it is important and necessary to understand the nature of the market and the interactions among that market's participants in trying to interpret any particular association and relationship.

However, to our knowledge, there might be alternative explanations for the co-movement of stocks in the market. It could be the general economic factors that drive the markets, and the stocks' upward or downward trend is based on general market sentiment (Vo 2014). For example, stock prices may be more synchronous in the boom or bust cycle as well as in recession or crisis time. The time that a single stock moves on its own, independently from the general market, is the time of earnings or news (whether good or bad) announcement. Beside poor information environment, noise could also result in a lower R-square, but this outcome can lead to different interpretations about the relationship between R-square and anomalies in the markets (Hu and Liu 2013). Therefore, we recognize that our

interpretations, though supported to some extent by the research's results, remain conjectures, and we invite alternative explanations of our findings.

**Author Contributions:** A.H.N., T.M.T.V., and Q.T.T.D. jointly conceived the research idea. A.H.N. did the literature review and hypotheses development. T.M.T.V. and Q.T.T.D. put together the dataset and research methodology. These authors contributed equally to carrying out the analysis as well as in writing the paper. All authors have read and agreed to the published version of the manuscript.

**Funding:** The research was funded by the National Economics University, Hanoi, Vietnam.

**Acknowledgments:** The authors would like to send our thanks the National Economics University (NEU), Vietnam for funding this research and anonymous reviewers for their supportive comments and suggestions.

**Conflicts of Interest:** The authors declare no conflict of interest.

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
