# Peer review of "Corporate Governance and Stock Price Synchronicity: Empirical Evidence from Vietnam"

_ijfs, doi:10.3390/ijfs8020022_

Round 1

Reviewer 1 Report

I see value in your study to examine the impact of corporate governance on stock price informativeness in the emerging   Vietnamese stock-market. However, for me to support your paper for publication I would want to see three main changes to the current manuscript:

1.your literature review  write-up and subsequent hypotheses is somewhat confusing as you switch between whether high or low stock price synchronicity implies better corporate governance. Unfortunately, you have provided an argument for it to go either way and so the interpretation that your variables of interest have on the direction of synchronicity is suspect. I suggest you rebuild your literature review and hypothesis to focus directly on what the emerging market literature says the direction of the relationship should be and limit it to that so that the reader can see that your hypothesis for your empirical work (which is effectively a replication study) is directly comparable to other papers.

2. You are lacking a good set of control variables. For example, one argument that you make is that better established firms with improved corporate governance will lead to higher synchronicity - if this is true then you should control for how well known and established the firm is. This can be done by including the age of the firm (such as by its IPO date) as well as controlling for the proportion of marketable shares there are in the firm (in other words how many shares are available to purchase versus those which are held by block holders). You should also control for the standard deviation of the firm's stock price (risk) and perhaps a couple of other things that you can gather information on. I am not familiar with the Vietnamese market and so do not know what additional information exists but I would assume you could at least calculate things such as the market to book  and leverage ratios? At the very least you need to work on a slightly expanded set of controls.

3. There is only one valid set of regression results that you are generating which are from the fixed effects model. Remove the results of all other regressions unless you have a valid reason, for example, to suspect the need for random effects (which i personally cannot see the econometric justification for). In other words, running a number of alternative econometric models does not make for a better paper if the models themselves are invalid!

Focusing on fixed effects, you need to also expand exactly what fixed effects you're looking at. Ideally, you wish to include both firm and year fixed effects. In addition, you should cluster the standard errors by the firm. At the very least they should be heteroscedastic consistent standard errors (White's correction). Doing what I have just suggested may have substantial implications on your results.

Something else you can do to support your hypotheses is conduct some univariate results. For example, split your firms into two subsamples based on their board size. Then compare the difference in stock price synchronicity between the subsample of firms that have low board size compared to those with high board size. The difference should be significant and in the direction that you hypothesize. Doing things like this can further support your claims as the results will be model-free.

Finally, I found the quality of the English grammar to generally be good but it still needs professional editing.

Good luck.

Author Response

Response to Reviewer 1 Comments

 Point 1: Your literature review write-up and subsequent hypotheses are somewhat confusing as you switch between whether high or low stock price synchronicity implies better corporate governance. Unfortunately, you have provided an argument for it to go either way and so the interpretation that your variables of interest have on the direction of synchronicity is suspect. I suggest you rebuild your literature review and hypothesis to focus directly on what the emerging market literature says the direction of the relationship should be and limit it to that so that the reader can see that your hypothesis for your empirical work (which is effectively a replication study) is directly comparable to other papers.

 Response to Point 1: We have rebuilt the literature review to make it more clear and easier to understand:

- Line 103 to 116: We added more explanations for the concept “information environment”.

- Line 116 to 119: The introduction of the view “Lower synchronicity is associated with a more efficient and transparent information environment” is shortened to only 4 lines in order to avoid misunderstanding and confusion with the other view supported by our research.

- Line 133 to 140: The paragraph is moved to this place to explain for the concept “Corporate Governance” mentioned inline 131.

- Line 284 and 299: We reworded the two hypotheses about the relationship between board size, board independence and stock price synchronicity to make them more consistent with the other two hypotheses.

Point 2: You are lacking a good set of control variables. For example, one argument that you make is that better-established firms with improved corporate governance will lead to higher synchronicity - if this is true then you should control for how well known and established the firm is. This can be done by including the age of the firm (such as by its IPO date) as well as controlling for the proportion of marketable shares there are in the firm (in other words how many shares are available to purchase versus those which are held by block holders). You should also control for the standard deviation of the firm's stock price (risk) and perhaps a couple of other things that you can gather information on. I am not familiar with the Vietnamese market and so do not know what additional information exists but I would assume you could at least calculate things such as the market to book and leverage ratios? At the very least you need to work on a slightly expanded set of controls.

Response to Point 2: We have expanded the model as follow (shown inline 360 to 364):

 - Four control variables are added into the model: SOWN (State ownership), AGE (Firm age) RISK (The standard deviation of stock prices) and LEV (Leverage).

- Due to the limit of data for the proportion of marketable shares of firms, we could not collect this information. However, the variable “State ownership” (SOWN) is added as an alternative as shares owned by the government are also restricted to be traded in the market.

- The market to book ratio was already included in the model with the name “Growth opportunity” and code “GROW”. We then change its name and code to “Market to book ratio” and “MTB” respectively for the direct reflection.  

Overall, the new model includes 4 independent variables and 7 control variables. And our new samples now, therefore, contains 243 listed companies with 1,121 observations. Accordingly, this leads to the changes of the results shown in Table 2 - Descriptive statistics (Line 434 to 435) and Table 3 – Correlations between variables (Line 475 to 476).

Point 3:

3.1. There is only one valid set of regression results that you are generating which are from the fixed-effects model. Remove the results of all other regressions unless you have a valid reason, for example, to suspect the need for random effects (which I personally cannot see the econometric justification for). In other words, running a number of alternative econometric models does not make for a better paper if the models themselves are invalid!

3.2. Focusing on fixed effects, you need to also expand exactly what fixed effects you're looking at. Ideally, you wish to include both firm and year fixed effects. In addition, you should cluster the standard errors by the firm. At the very least they should be heteroscedastic consistent standard errors (White's correction). Doing what I have just suggested may have substantial implications on your results.

3.3. Something else you can do to support your hypotheses is to conduct some univariate results. For example, split your firms into two subsamples based on their board size. Then compare the difference in stock price synchronicity between the subsample of firms that have a low board size compared to those with high board size. The difference should be significant and in the direction that you hypothesize. Doing things like this can further support your claims as the results will be model-free.

Response to Point 3:

 3.1. We have removed the results of Pooled OLS and random effect models. Table 5 (from line 530 to 531) shows the results for a fixed-effect model (FEM) and robust FEM only.

 3.2. In the fixed-effect model, we included a firm fixed effect and detected the problem of heteroskedasticity in the model. Therefore, to correct the problem, we cluster (robust) the standard errors by the firm as advised by the reviewer. The description and results are shown from line 508 to 513 and in table 5 (from line 530 to 531).

3.3. For each independent variable, we divided the firms into 2 subgroups based on the variable average value to observe the differences in R-square. The descriptive statistics and interpretations are shown from line 477 to 485.

- In addition, there are some other main changes made by the authors as follow:

+ We have added some points about the efficiency of the market to the introduction. Firstly, we relate the concept of the efficient market with stock price synchronicity and information environment especially in the context of an emerging market (from lines 28 to 30 and from lines 40 to 54). Secondly, we briefly review the literature on the improvement of the market efficiency in Vietnam through the period of time (from line 57 to 66). That is to say Vietnam market had transformed from an inefficient form to be efficient in the weak-form accompanied by the characteristics of the emerging market which, therefore, creates a general background for the research.

+ In conclusion, we decrease the strongness of the conclusion by providing some more alternative explanations for our findings and recognizing that our interpretation remains conjecture (From line 585 to 594).

We would like to send our thanks to the reviewer for your valuable recommendations and suggestions to improve our paper.

Reviewer 2 Report

I think the work is well executed and the idea is interesting. Nevertheless, I feel, that the concept of the stock market synchronicity is a bit "worrying" in the context of the less developed Vietnamese market. It will be good to address the question of marker efficiency before embarking upon a discussion of the co-movements that is supposed to be to the feature of increased corporate governance. in the context of Ho Chi Minh Stock Exchange (HOSE).
I suggest that there may be another reason, why shares in Vietnam move together. One example may be the phase of the economic cycle in general. I think that if the market is underdeveloped it sounds quite courageous to make any conclusions about influence corporate governance.
This is to say that share and market instruments tend to move together. this effect does not have to be caused by inner characteristics of individual companies listed on a stock exchange. It can be caused by the general economic environment and it can be underlined by the particular phase of the business cycle. Like boom and bust cycle. I feel that it would be worth exploring the issue of market efficiency of Ho Chi Minh Stock Exchange (HOSE) before these tests. I think that it is quite a challenge to make such strong conclusions about the influence of corporate governance aspects in the underdeveloped markets and to arrive at so strong conclusions

I would like to encourage that authors think bout this aspect and inclúde it in their reasoning and analysis.

Author Response

Response to Reviewer 2 Comments

Point 1: I think the work is well executed and the idea is interesting. Nevertheless, I feel, that the concept of the stock market synchronicity is a bit "worrying" in the context of the less developed Vietnamese market. It will be good to address the question of marker efficiency before embarking upon a discussion of the co-movements that is supposed to be to the feature of increased corporate governance in the context of Ho Chi Minh Stock Exchange (HOSE).
I suggest that there may be another reason, why shares in Vietnam move together. One example may be the phase of the economic cycle in general. I think that if the market is underdeveloped it sounds quite courageous to make any conclusions about influence corporate governance.

This is to say that share and market instruments tend to move together. This effect does not have to be caused by the inner characteristics of individual companies listed on a stock exchange. It can be caused by the general economic environment and it can be underlined by the particular phase of the business cycle. Like boom and bust cycle. I feel that it would be worth exploring the issue of the market efficiency of Ho Chi Minh Stock Exchange (HOSE) before these tests. I think that it is quite a challenge to make such strong conclusions about the influence of corporate governance aspects in the underdeveloped markets and to arrive at so strong conclusions

I would like to encourage that authors think bout this aspect and include it in their reasoning and analysis.

Response to Point 1:

- We have added some points about the efficiency of the market to the introduction. Firstly, we relate the concept of the efficient market with stock price synchronicity and information environment especially in the context of an emerging market (from lines 28 to 30 and from lines 40 to 54). Secondly, we briefly review the literature on the improvement of the market efficiency in Vietnam through the period of time (from line 57 to 66). That is to say Vietnam market had transformed from an inefficient form to be efficient in the weak-form accompanied by the characteristics of the emerging market which, therefore, creates a general background for the research.

- In conclusion, we decrease the strongness of the conclusion by providing some more alternative explanations for our findings and recognizing that our interpretation remains conjecture (From line 585 to 594).

- In addition, there are some other main changes made by the authors as follow:

+ In the literature review, the introduction of the view “lower synchronicity is associated with a more efficient and transparent information environment” is shortened to only 4 lines in order to avoid misunderstanding and confusion with the other vỉew supported by our research (From line 116 to 119).

+ We reworded the two hypotheses about the relationship between board size, board independence, and stock price synchronicity to make them more consistent with the other two hypotheses (Line 284 and line 299).

+ We added four new control variables to our model including SOWN (State ownership), AGE (Firm age), RISK (The standard deviation of stock prices) and LEV (Leverage) and change the name of the variable “Grow opportunity” (coded as “GROW”) to “Market to book ratio” (coded as “MTB”) (From line 360 to 364).

+ We have removed the results of OLS and random effect models as they are invalid. Table 5 (line 530 to 531) shows the results for only a fixed-effect model (FEM) and robust FEM. In the fixed-effect model, we include firm fixed effect and detect the problem of heteroskedasticity in the model. Therefore, to correct the problem, we cluster (robust) the standard errors by the firm as advised by the reviewer. The description and results are shown from line 508 to 513 and in table 5 (from line 530 to 531

We would like to send our thanks to the reviewer for your valuable recommendations and suggestions to improve our paper.

Round 2

Reviewer 1 Report

The paper has improved and I appreciate you have improved your model. However, you need to change your abstract as you still refer to using Random effects, when now you have deleted it. please update your abstract.

i recommend an English proofread check your work as well. The language quality is good, but not perfect.

Author Response

Point 1: The paper has improved and I appreciate you have improved your model. However, you need to change your abstract as you still refer to using Random effects, when now you have deleted it. Please update your abstract.

I recommend an English proofread to check for your work as well. The language quality is good, but not perfect.

Response 1:

- We have updated the overview of the research method used in the study and also re-arranged a few sentences in the abstract of the research (From line 9 to line 15).

- We did the English Proofread check with the supports of our native English speaker colleague and English proofreading tools to correct our errors regarding grammar, punctuation, spelling, word choice, etc.,

- Besides, in the introduction, the difference in information environment between the developed market (efficient market) and the emerging market (less efficient market) is explained in detail to make it clearer why investigating the relation between information environment and stock price synchronicity in the emerging market is interesting for the researchers. (From line 43 to line 71)

- Also, more discussions are added to interpret the relationship between foreign ownership, managerial ownership and stock price synchronicity to make the results more revealing. (From line 602 to line 608 and from line 620 to 627)

Once again, we would like to express our appreciation to the reviewer for your supportive recommendations and suggestions.

Reviewer 2 Report

To assess the changes in the manuscript the authors made stronger links between the stock price synchronicity and efficiency of the market and highlighted the validity of this conjunction especially in the context of the information environment of an emerging market in Vietnam,

There is a more thorough description of the improvement of the market efficiency in Vietnam through the period of time.  The authors have added information about efficiency testing on Vietnamese market including relevant literature.

As to the models on Synchronicity with the addition of new control variables including SOWN (State ownership), AGE (Firm age), RISK (The standard deviation of stock prices) and LEV (Leverage) renaming the variable “Grow opportunity” (coded as “GROW”) to “Market to book ratio”  the model is more intuitive and can be better understand. The results are more revealing and more revealing

I appreciate that the authors have introduced new hypotheses and recognized the importance of market efficiency linked to share co-movements.

I have read carefully the amendments to the paper and I think that the paper is now more realistic and it can now be published.

Author Response

Point 1: In the evaluating chart, the reviewer expects that the introduction and result interpretations will be improved. So we continue to make some more amendments.

Response 1:

- The difference in information environment between the developed market (efficient market) and the emerging market (less efficient market) is explained in detail to make it clearer why investigating the relation between information environment and stock price synchronicity in the emerging market is interesting for the researchers. (From line 43 to line 71)

- We added more discussions interpreting the relationship between foreign ownership, managerial ownership, and stock price synchronicity to make the results more revealing. (From line 602 to line 608 and from line 620 to 627)

- Also, we have updated the overview of the research method used in the study and also re-arranged a few sentences in the abstract of the research (From line 9 to line 15).

- In addition, we did the English Proofread check with the supports of our native English speaker colleague and English proofreading tools to correct our errors regarding grammar, punctuation, spelling, word choice, etc.

Once again, we would like to express our appreciation to the reviewer for your supportive recommendations and suggestions.